# Towards Establishing Cross-Platform Interoperability for Sensors in Smart Cities

**DOI:** 10.3390/s19030562

**Published:** 2019-01-29

**Authors:** Kanishk Chaturvedi, Thomas H. Kolbe

**Affiliations:** Chair of Geoinformatics, Technical University of Munich, Arcisstrasse 21, 80333 Munich, Germany; thomas.kolbe@tum.de

**Keywords:** smart cities, sensors, Internet of Things, interoperability, sensor web enablement, OGC standards

## Abstract

Typically, smart city projects involve complex distributed systems having multiple stakeholders and diverse applications. These applications involve a multitude of sensor and IoT platforms for managing different types of timeseries observations. In many scenarios, timeseries data is the result of specific simulations and is stored in databases and even simple files. To make well-informed decisions, it is essential to have a proper data integration strategy, which must allow working with heterogeneous data sources and platforms in interoperable ways. In this paper, we present a new lightweight web service called InterSensor Service allowing to simply connect to multiple IoT platforms, simulation specific data, databases, and simple files and retrieving their observations without worrying about data storage and the multitude of different APIs. The service encodes these observations “on-the-fly” according to the standardized external interfaces such as the OGC Sensor Observation Service and OGC SensorThings API. In this way, the heterogeneous observations can be analyzed and visualized in a unified way. The service can be deployed not only by the users to connect to different sources but also by providers and stakeholders to simply add further interfaces to their platforms realizing interoperability according to international standards. We have developed a Java-based implementation of the InterSensor Service, which is being offered free as open source software. The service is already being used in smart city projects and one application for the district Queen Elizabeth Olympic Park in London is shown in this paper.

## 1. Introduction and Motivation

With the rapidly increasing urban population, it is essential for local governments to efficiently manage the city’s resources, development, and operation. Smart Cities is an emerging field in the same direction that *"relies on advanced data processing with the goals of making city governance more efficient, citizens happier, businesses more prosperous and the environment more sustainable"* [1]. It allows managing city resources such as energy and water with the help of advanced information and communication technologies such as Sensors and the Internet of Things (IoT) [2], Big Data [3], Cloud Computing [4], and also geospatial technologies [5]. Ubiquitous sensors and IoT devices are essential parts of several smart infrastructures providing detailed information by sensing the environment. Many application domains, such as home and industrial automation, intelligent energy management and smart grids, traffic management, and others benefit from the use of real-time sensor observations. These sensors can be stationary such as smart meters [6] and weather stations [7]. Some of the sensors can also be non-stationary such as moving sensors for measuring air quality [8]. We believe that there is also another category of virtual sensors which are not necessarily located physically, but their sensing observations can be studied to get better information about our surroundings and environment. For example, real-time social media analytics like Twitter feeds can be used for behavioral and sentiment analysis and to make better decisions [9].

The Smart City concept is emerging rapidly and many cities worldwide are developing their smart infrastructures. Commercial implementations include IBM Smarter Planet [10], CityNext [11] from Microsoft, and The Internet of Everything for Cities [12] by CISCO. Some of the projects are also run by consortia of universities, companies and city councils in a collaborative manner such as Smart Sustainable Districts [13], under Climate-KIC of the European Institute of Innovation & Technology (EIT) and the project EU ICT 30-2015 (Internet of Things and Platforms for Connected Smart Objects) (http://ec.europa.eu/research/participants/portal/desktop/en/opportunities/h2020/topics/ict-30-2015.html) funded by the European Union Horizon 2020 Programme. In most smart city projects, multiple stakeholders and companies are involved who may be the owners, operators, utility companies, sensor providers, citizens, and visitors. These stakeholders usually are interested in specific applications or simulations and collect data for their own purposes. For example, an energy provider company participating in a project owns the energy consumption data for buildings. In general scenarios, this data is meant to be used with the application owned by the same energy provider company. Most often, the structure of the data is not standardized and lacks explicit semantics, and hence, is not suitable to work with other datasets. This is typically also the case for sensor and IoT platforms being used in such projects. In most scenarios, stakeholders use their own sensors which are built for specific purposes and are based on specific platforms (c.f. Section 2.2). These platforms may be open or proprietary, however, most of the time, they are not standardized. Another challenge is that the APIs associated with these platforms are subsequently changed without notifying the users. Moreover, the observations retrieved from these sensors are not always associated with an API. In many scenarios, such timeseries observations are the results of simulations [14] which are stored in databases or even simple files (c.f. Section 2.3). This leads to a major challenge to work in unified ways with a wide variety of data sources and their data types which are completely different from each other.

It shows that such smart city projects involve complex distributed systems having multiple stakeholders, diverse applications, a multitude of sensor and IoT platforms and data sources. To make well-informed decisions, it is very important to achieve a proper data integration strategy, which must allow working with heterogeneous data sources and platforms in a common operational framework. Such integrated information leads to joint and real-time analytics to manage aspects of how a city functions and is managed e.g., by using smart city dashboards [15] as shown in Figure 1. However, as highlighted by Moshrefzadeh et al. [16], due to data privacy concerns and competition between several stakeholders, it does not make sense to try to collect all available data resources within a central data repository. Rather, the data should remain with their owners and should be combined flexibly according to specific applications or stakeholders. This leads to the requirement of interoperability in order to deal with the heterogeneous sensor and IoT platforms. Such interoperability can be achieved by using open and international standards which, on the one hand, allow modeling and representing the data sources and, on the other hand, allow interfacing the distributed components that give access to data, visualizations, and analytical tools.

Sensor Web Enablement (SWE) [17], an initiative from the Open Geospatial Consortium (OGC), has already developed a suite of standards enabling the discovery, access, tasking, as well as eventing and alerting of the sensor resources in a standardized way. The OGC SWE standards suite comprises well-defined information models such as (i) SensorML [18], which not only represents sensor description and metadata, but also sensor calibration records and accuracy and precision information, and (ii) Observations and Measurements (O&M) [19] for describing real-time sensor observations. The SWE also provides comprehensive interface models and web services such as Sensor Observation Service (SOS) [20] and SensorThings API [21] for retrieval of sensor descriptions and observations with the help of standardized requests. In comparison to SOS, SensorThings API is a relatively new standard, which is REST-ful, lightweight, and based on JSON. The Timeseries API [22] developed by a company called 52° North is not an international standard, however, provides a REST-ful web binding to the OGC Sensor Observation Service in order to be easily queried and visualized by lightweight web applications. Other than OGC SWE, there are also several projects such as bIoTope [23], VICINITY [24], BIG IoT [25], and FIWARE [26] (more details in Section 2) dealing with interoperability issues over heterogeneous sensor and IoT devices in the Smart Cities domain.

Such sensor web infrastructures play an important role in establishing interoperability for heterogeneous sensors and are considered as one of the keys to work in distributed scenarios. They allow encoding sensor description and observations using well-defined standards as well as accessing them using standardized interfaces. In this way, applications and tools can be developed based on these standards without worrying about what different kinds of sensors they use. Multiple sensors can be attached to these infrastructures and their interfaces will always be common for different applications. There are several projects which are already realizing such sensor web infrastructures (c.f. Section 2.1). However, such infrastructures always require a data storage to store sensor metadata and their observations, based on which web services can query and retrieve data and observations. The issue is that in a distributed environment, where multiple stakeholders and sensor owners are involved with proprietary sensors, not all of them would be willing to inject their proprietary data into a third-party data storage in the sensor web. Moreover, in a running distributed system having another data storage for the sensor web will require regular maintenance. It can also be a complex affair while moving the infrastructure to different locations, for example, from one server to another or into the cloud. Additionally, not many standardized sensor web infrastructures yet consider supporting virtual sensors such as information coming from Twitter feeds or simple files. In these cases, it is needed to have an intermediate service which can connect to a specific data source and encodes the observations “on-the-fly” according to the standardized interfaces without worrying about the data storage and multitude of data sources (see Figure 2). In other words, this intermediate service should be like a “Babel Fish” from the Hitchhiker’s Guide to the Galaxy [27] which is a *"universal translator that neatly crosses the language divide between any species"*.

This paper provides solutions for the above-mentioned issues by introducing the lightweight InterSensor Service [28]. This service provides several data adapters which can be used for establishing connections to not only different IoT platforms, but also to external databases, CSV files, Cloud-based spreadsheets, GPS feeds, and real-time Twitter feeds. While querying, the service opens a data source connection and retrieves the observations based on querying parameters directly from the data source. The service encodes these observations “on-the-fly” according to the international standardized interfaces such as the OGC Sensor Observation Service and OGC SensorThings API. In this way, applications compliant to such OGC standardized interfaces can simply be used to interact with heterogeneous observations without worrying about their data storage. The major reasons for initial development for the responses according to the OGC SWE interfaces are as follows. First, the OGC SWE framework is completely based on released and published Open Standards adopted internationally. When implementing something that is not standardized, there is a high risk that the developed and suggested encodings/APIs will be abandoned, replaced, or vanish after the project is over. Second, in Smart Cities, a lot of other data and presentation services such as web maps, 3D visualizations, data with geographic coverages like weather data, air quality, wind fields etc. are provided by Spatial Data Infrastructures (SDIs). All of these services are also provided using OGC standards. Hence, Sensor and IoT services just add another category of web service to SDIs and it is beneficial to make the IoT service compliant to SDIs such that they can be used with similar protocols and tools already used in the framework of SDIs. Third, it also makes the observations suitable to be visualized and managed with the other numerous OGC geospatial standards such as CityGML [29]. Also, no other implementation yet provides such “on-the-fly” interfaces for international OGC SWE standards. However, the concept is not limited to only the OGC standards. In the future, interfaces can also be developed according to other standards/protocols such as FIWARE. The InterSensor Service is a Java-based application and is available for free as Open Source software [30].

This paper is a substantially extended version of earlier work presented at the IEEE International Smart Cities Conference 2018 [28] and provides a comprehensive analysis of the InterSensor Service by comparing it with other research work and by implementing data adapters and interfaces for supporting more use cases. The paper also discusses deployment options for different distributed scenarios considering the security and privacy of the data sources and platforms. Furthermore, the paper presents a first evaluation of the InterSensor Service by comparing original response times and data payload sizes against “on-the-fly” data and protocol conversion using InterSensor Service when connecting to different data sources. The rest of this paper is structured as follows: Section 2 gives a comprehensive literature review of existing sensor web infrastructures and IoT platforms being used in smart city initiatives. This section also reviews different possibilities for storing and managing observations, which can be retrieved by the InterSensor Service. The architecture and details of the data model of the service are described in Section 3. Section 4 demonstrates the details of implementing and using the InterSensor service by giving configuration examples to connect to individual data sources. This section also shows the details of standardized interfaces generated automatically by the service. Section 5 shows demonstration scenarios of the InterSensor Service being already used in a smart city project based in the Queen Elizabeth Olympic Park in London. The last section draws the conclusions about the presented work and outlines the relevant aspects of our future research and development tasks.

## 2. Literature Review

Depending on the use cases and sensor hardware, there are numerous possibilities for managing sensor observations. These observations can be stored and managed using different platforms and APIs, databases, and simple external files as shown in Table 1.

### 2.1. Smart Cities, Sensors and Interoperability

Owing to the well-defined and comprehensive set of open and international standards, the Sensor Web Enablement (SWE) standard suite is already being used worldwide in various domains such as early warning systems [56], disaster management [57], marine science [58,59], citizen science [60], environmental and air quality monitoring [61,62] and many more. Smart City initiatives are also recognizing the importance of such sensor web infrastructures. The “Smart Cities Spatial Information Framework” [63] is based on the integration of OGC open standards and geospatial technology and is critical achieving the benefits of spatial communication for smart cities. The OGC Innovation Program [64] also includes testbeds and pilots for smart city infrastructures. One such initiative recognizing the importance of such open and interoperable standards recently completed its first phase in Europe: OGC’s Future City Pilot [65]. The Future City Pilot Phase 1 (FCP1) is an OGC Interoperability Program initiative in collaboration with buildingSMART International (bSI). The pilot aimed at demonstrating and enhancing the ability of spatial data infrastructures to support quality of life, civic initiatives, and urban resilience. One of the objectives of the pilot was to demonstrate “how dynamic city models can provide better services to the citizens as well as can help to perform the better analysis?”. Within this use case, the city’s static data such as buildings or houses with elderly citizens having special needs could be integrated with dynamic data such as outside temperature or air humidity using interoperable OGC standards such as CityGML [29] and Sensor Observation Service [20]. Such potential integration within council owned assets could lead to better decision making in case of extreme weather or other emergency scenarios matching human needs to the right housing or resources. Another initiative is the ESPRESSO project [66] aiming to provide cities and communities the ways for implementing enhanced interoperable and standards-based architecture for their specific city contexts. This project defines key elements and concepts required to be addressed to achieve interoperability between various services within a city and also to increase the interoperability between different cities. The concepts developed under this project have already been tested and proven in Rotterdam (the Netherlands) and Tartu (Estonia). For managing heterogeneous resources within complex distributed systems, Moshrefzadeh et al. [16] propose a new concept called Smart District Data Infrastructure (SDDI). This framework allows integrating diverse components such as stakeholders, sensors, IoT devices and simulation tools with a virtual district model representing the physical reality of the district. To access distributed resources, the framework uses a well-defined set of OGC-based service interfaces such as Web Feature Service [67], Web Coverage Service [68], Catalog Service for the Web [69], Sensor Observation Service [20] and SensorThings API [21]. There are also several projects and frameworks using OGC-based standards in smart city contexts such as Smart Cities Intelligence System (SMACiSYS) [70], MONICA project [71], i_city project for visualizing e-bike usages [72], and Smart Emission project [38]. The open source implementations such as 52° North Sensor Observation Service [73] and the FRaunhofer Opensource SensorThings (FROST) Server [74] allow inserting, querying, and visualizing arbitrary sensor data and observations according to the OGC Sensor Observation Service and OGC SensorThings API standards respectively.

Apart from OGC Sensor Web Enablement, there are also other architectures and frameworks which focus on interoperability of sensor and IoT devices and being applied in different projects. The FIWARE [26] is a generic and open-source platform that aims to make interoperable city services, to provide access to real-time context information, and to implement smart city applications. The platform enables developers and communities to create their services based on commonly defined APIs and data models. The FIWARE is already being used in several smart city initiatives such as “City Enabler” [75]. It is a FIWARE-based software product allowing scattered and distributed urban data to be collected and organized in a central repository, which can be fed to different applications with the help of the standard APIs. Other than FIWARE, another project called BIG IoT [25] focuses on cross-standard, cross-platform, and cross-domain IoT services and applications. The approach is to register an individual IoT platform to their so-called “BIG-IoT Marketplace”, which acts as a catalog. Using the Marketplace, the BIG-IoT API allows discovering, authenticating/authorizing multiple IoT resources and allows using them in a single application. Similarly, the bIoTope project [23] under the European Union’s Horizon 2020 Programme provides an ecosystem allowing registering heterogeneous IoT platforms and accessing them using standardized and open APIs. Like BIG-IoT Marketplace, the bIoTope project also includes a Marketplace called IoTBnB which can be used for discovering and authenticating the different IoT platforms. In the similar ways, another EU Horizon 2020 project VICINITY [24] provides a decentralized ecosystem offering “interoperability as a service”. Its architecture involves a VICINITY Cloud acting as a Marketplace used for registering and then discovering and accessing the numerous IoT platforms using the standardized APIs. Other pertinent initiatives carried out within the EU Horizon 2020 Programme are symbIoTe [35], INTER-IoT [36], and Thingful [37]. Another interesting initiative is Smart Emission Data Infrastructure [38] which includes the use of international open standards to achieve interoperability and provide open access to the sensor data. It involves a centralized repository where the raw sensor data is harvested and using ETL processes, the data is published according to the OGC Sensor Observation Service, SensorThings API, and FIWARE. Sensor Measurement Lists (SenML) [39] is also a specification working towards interoperability of sensors. In this specification, representations share a common SenML data model. A simple sensor, such as a temperature sensor, could use this media type in protocols such as HTTP to transport the measurements of the sensor or to be configured. Jazayeri et al. [76] also provide a comprehensive evaluation of four open interoperable standards for the IoT devices: OGC PUCK over Bluetooth, TinySOS, SOS over CoAP, and OGC SensorThings API.

The importance of open and interoperable solutions for smart cities is also being recognized in the form of developing user guides for cities and stakeholders and by organizing hackathons and webinars for encouraging innovative application ideas. The Smart City Interoperability Reference Architecture (SCIRA) project [77] is an initiative by the OGC Innovation Program. The purpose of this project is to advance standards for Smart Safe Cities and develop open and interoperable designs for incorporating IoT sensors into city services. The project aims to provide free deployment guides, reusable design patterns, and other resources that municipalities can use to plan and implement standards-based Smart City systems using technologies such as IoT, Sensor Webs, and Geospatial Frameworks. As part of SCIRA, a hackathon “Hacks and the City” (https://scira.ogc.org/hack) has been organized which encouraged participants to design and implement new application ideas that use a variety of city datasets and data sources to improve public safety, responder awareness, and community resilience. Similarly, another hackathon “Neue Wege für die Mobilität in Augsburg” (www.neue-wege-augsburg.de) also aims to develop new ideas and solutions for mobility by using a data platform based on open and international OGC-based standards. The project Enabling Smarter Cities [78] initiated by EIT Climate-KIC organizes a series of webinars intending to provide key decision-makers with awareness on the importance and needs of interoperable smart city solutions.

As mentioned, there are several projects, initiatives, and frameworks which are dealing with interoperability of sensor and IoT observations in their own ways. However, the implementation of all of the approaches always requires a data storage (e.g., a database repository) to store sensor metadata and their observations. This data storage allows interfaces and web services to query and retrieve sensor data and observations. The issue is that in a distributed environment, where multiple stakeholders and sensor owners are involved with proprietary sensors, not all of them would be willing to inject their proprietary data into such third-party data storage in the sensor web. Moreover, in a running distributed system having another data storage for the sensor web will require regular maintenance. It can also be a complex affair while moving the infrastructure to different locations, for example, from one server to another or into the cloud.

### 2.2. Different Sensor and IoT Platforms

There are several platforms which consist of a complete suite for managing sensor observations. These platforms include their own data storage, visualization clients, and the APIs to query and retrieve the observations. ThingSpeak [31] is an IoT platform that allows users to register different sensors attached to simple microcontrollers such as Arduino and Raspberry Pi, collect and store sensor observations in the Cloud and develop IoT applications. The ThingSpeak platform provides applications to analyze and visualize observations. The system also allows querying by location, allowing the user to have access to data from various locations in the world. OpenSensors [32] which is termed as “Twitter for Sensors” [79] allows users to connect diverse sensor devices and publish their observations for free. The data is publicly accessible, shareable and reusable by and for anyone. The platform provides real-time and historical access to public and private data through the API and in-browser data view. The Things Network [33] is a relatively new initiative aiming at building a network for the Internet of Things by creating abundant data connectivity. The network focuses on a technology called LoRaWAN [80] which allows for things to talk to the Internet without 3G or WiFi, so no WiFi codes and mobile subscriptions are required. It features low battery usage, long range and low bandwidth, which is ideal for the IoT devices. The Things Network also supports publishing observations to other platforms such as OpenSensors [81] and OGC Sensor Observation Service [82]. Such integration makes discovering, analyzing, and visualizing sensor observations even easier. The weather has a major influence on city systems ranging from energy and water, to sanitation, transportation, health care, to disaster management. Weather Underground [34] is a commercial weather service providing real-time weather information via the Internet. It provides weather reports for most major cities across the world on its website. It also uses observations from members with automated personal weather stations (PWS). Weather Underground currently uses observations from over 250,000 personal weather stations worldwide.

Likewise, there may even be more platforms being used in smart city projects. Such platforms may be open or proprietary in nature, however, most often, their associated APIs are not publicly documented. Another challenge is that these APIs are subsequently changed without notifying the users. Also, these platforms are meant for different purposes. For example, Weather Underground is used for weather stations while Thingspeak can be used for attaching an indoor DHT22 sensor. The completely different APIs for such platforms make it difficult for end-user applications to analyze and visualize them together. One possibility is to attach these multiple platforms to the OGC Sensor Web Enablement-based implementations such as 52° North SOS and FROST Server in order to analyze them using common interfaces. However, such implementations require importing the observations from the original platform and storing them in their own data storages. The issue with such an approach is that it leads to data redundancy. The respective platform such as Thingspeak already stores observations in its own data storage. Another challenge is that in some cases, these platforms may also be proprietary. In this case, the owners would like to avoid storing their proprietary data to a third-party data storage.

### 2.3. Other Sources of Timeseries Data

The timeseries data retrieved from the sensors and IoT devices are not always associated with an API. In many scenarios, such timeseries data are also stored in databases. Traditional relational database management systems such as Oracle [40], MySQL [41], and PostgreSQL [42] are already being used with many sensor platforms such as the 52° North Sensor Observation Service implementation [73] and the FROST server implementation for the SensorThings API [74]. They provide standard SQL functions to query and analyze sensor data. In most scenarios, sensors continuously produce a huge amount of time series data, which creates a demand for efficient time series data analysis. TimescaleDB [43] and InfluxDB [44] are good examples of Open Source timeseries databases which are being used in the fields of IoT and real-time analytics. When it comes to managing more heterogeneous data generated by millions of sensors, devices and gateways, each with their own data structures, databases require new levels of flexibility, agility, and scalability. In this environment, NoSQL databases such as MongoDB [45] are proving their value. Another new concept in this direction is the Data Stream Management System (DSMS). Such management systems continuously process arriving data without having to persist them, this speeds up the data evaluation process, achieving more timely results in comparison to traditional DBMSs. Anjos et al. [83] explore the feasibility of Data Stream Management Systems (DSMSs) to support Energy Management applications, pointing out how to implement an Energy Management System capable of real-time data processing.

In many scenarios, especially, when observations are not very highly frequent, time-varying data are stored in external files such as Comma Separated Values (CSV) and Excel sheets. Such files are usually generated once for a specific scenario and do not update continuously. There are also cloud-based systems such as Google Fusion Table [46], Google Spreadsheet [47], and Microsoft OneDrive [48] which allow users to store such time series data in a cloud environment. Applications such as traffic simulations and navigation involve locations which change with respect to time. Such moving objects can be stored using different file formats such as GPS Exchange Format (GPX) [49], Keyhole Markup Language (KML) [50] and Cesium Language (CZML) [51]. There are also platforms such as Waze API [52] allowing accessing real-time crowd-sourced traffic information using API requests.

In the field of semantic 3D city models, ongoing research allows supporting time-dependent and dynamic data from sensors and simulations with the city objects. CityGML [29], which is an international OGC standard to represent semantic 3D city models, is going to have a new module called Dynamizer [55]. It allows, on the one hand, representing time-dependent data and sensor data in standardized ways within a CityGML file, and on the other hand, providing a method for injecting dynamic variations of city object properties (like the electricity consumption of a building or the traffic density within a road segment) into the static representations. This concept has already been implemented within the initiative OGC Future City Pilot Phase 1 to support time-dependent solar power potential simulation results and to link real-time observations from Sensor Observation Services with 3D city objects [84,85].

Currently, there is no Sensor Web Enablement implementation which covers such data sources for retrieving timeseries data. The 52° North SOS Implementation and FROST Server support importing the timeseries data from a CSV file, however, the data is first imported to their data storage.

## 3. Cross-Platform Interoperability Using the InterSensor Service

As we learned in the previous section, there are numerous sources which may be used for accessing sensor information ranging from a multitude of platforms, databases, and simple files. For managing such heterogeneous data sources, we have developed the InterSensor Service, which is a very basic and lightweight web service. It allows users to connect to different data sources (as mentioned in Section 2) and retrieve their time-varying observations directly from the source without requiring any additional data storage. The simplified structure of the service allows linking heterogeneous sensor observations to be analyzed and visualized together. Additionally, the service also allows encoding observations according to international standardized interfaces such as the OGC Sensor Observation Service and OGC SensorThings API. We have also developed an interface for the 52° North Timeseries API. Although it is not an international standard, it provides a REST-ful web binding to the OGC Sensor Observation Service in order to be easily queried and visualized by lightweight web applications. In this way, different observations from heterogeneous data sources can be accessed by common applications compliant to these OGC SWE specifications.

### 3.1. Architecture

As shown in Figure 3, the architecture comprises of three layers.

#### 3.1.1. Data Adapters

This layer is responsible for establishing the connection to multiple data sources. The data sources can be (i) existing sensor and IoT platforms such as Thingspeak, OpenSensors, The Things Network, and OGC SWE standards, (ii) running databases such as Oracle, PostgreSQL, and TimescaleDB, and (iii) any external files located on a local machine, server, or cloud such as CSV and Excel sheets, GPX file (or GPS feeds embedded in a KML or CZML file), Google Fusion Tables, and CityGML Dynamizer files.

#### 3.1.2. Standardized External Interfaces

This layer is responsible for encoding queried observations from data sources according to well-defined interfaces. These interfaces include international standards such as the OGC Sensor Observation Service (SOS). The SOS interface allows querying data using operations such as *DescribeSensor* to retrieve sensor metadata according to the SensorML standard and *GetObservation* to retrieve sensor observations according to the O&M format. Another interface is Timeseries API [22], which is a RESTful web binding to the OGC Sensor Observation Service. It allows querying and visualizing sensor locations and real-time observations using the so-called Helgoland web client [86]. Similarly, observations can also be encoded and queried according to the SensorThings API interface.

#### 3.1.3. InterSensor Service

This is an intermediate layer which acts as a “Babel Fish” between the data sources and the interfaces. This layer is responsible for establishing connections to the individual data sources using adapters. After a successful connection, the service provides resources according to the data model (c.f. Section 3.2), which allows querying sensor observations and metadata using specified filters. The observations are mapped to the relevant resources in this layer. Furthermore, multiple interfaces can read the observations from this layer and encodes the data according to the desired interface. In this way, InterSensor Service, on the one hand, can query observations from heterogeneous and distributed data sources and map them using common and simple objects, and on the other hand, encodes them using standardized interfaces in order to analyze and visualize them together in a unified way.

### 3.2. Data Model

The InterSensor Service defines a few classes to connect to individual data sources. These classes contain specific attributes which can be used to connect to a particular data source. After successful connection to the data sources, the InterSensor Service forms three resources named *DataSource*, *Timeseries*, and *Observation* as shown in Figure 4. *DataSource* contains all the details of a specific data source whose link can be established using *DataSourceConnection*.

The details of each class are mentioned as follows.

#### 3.2.1. DataSourceConnection

As shown in Figure 5, this class allows users to specify parameters to connect to individual data sources. It contains metadata attributes such as name and description of the data source, what type of connection it is (e.g., a CSV file, JDBC connection, a web service etc.). Furthermore, it contains subclasses to connect to different resources. *ExternalFilesConnection* provides connection details to external files such as CSV, GPX, KML and CZML, and also to Cloud-based documents such as a Google Fusion Table. *DatabaseConnection* contains parameters to connect to a specific database. In similar ways, InterSensor Service can also be used to connect to CityGML Dynamizers using *DynamizerConnection*.

*PlatformConnection* is designed for connecting to different sensor and IoT platforms. This class has further subclasses for each platform, for example, *ThingSpeak*, *OpenSensors*, *OGC SensorThings*, *OGC Sensor Observation Service*, and *Twitter*. Each subclass contains specific properties for the connection to be established. For example, in case of the SensorThings API, *ThingId* is a unique ID to determine the details and metadata of a “Thing” (e.g., a weather station) such as https://example.sensorup.com/v1.0/Things(8774755). One “Thing” can deliver different observations (e.g., temperature, humidity etc.). Each observation can be determined by a *DatastreamId* such as https://example.sensorup.com/v1.0/Datastreams(8774757).

Hence, in order to add a timeseries property from the above mentioned SensorThings stream, the minimal inputs required will be a *baseURL* such as https://example.sensorup.com/v1.0, *ThingId* such as *8774755* and *DatastreamId* such as *8774757*. Similarly, a valid Thingspeak channel consists of a *baseURL*, a *channelID*, and *fieldID*. By providing these details, the InterSensor Service generates valid request calls and establishes a connection to the Thingspeak channel.

#### 3.2.2. DataSource

After providing details for the data source connection, the InterSensor Service validates the connection. Upon successful connection to the data source, it instantiates three resources. *DataSource* creates a unique ID for the data source and contains the details of *DataSourceConnection*. It also contains a list of available timeseries associated with it.

#### 3.2.3. Timeseries

Each *DataSource* can have multiple *Timeseries*. For example, if a data source is a running Thingspeak channel with two timeseries associated with it: temperature and humidity. In this case, the InterSensor Service creates two timeseries (one for temperature and the other for humidity) with two unique timeseries IDs associated with a common datasource ID. However, as per requirements, it is also possible to establish a connection to a specific timeseries from a data source connection (for example, only to the temperature stream).

#### 3.2.4. Observations

Both *DataSource* and *Timeseries* classes contain properties to connect to the data source. By providing querying filters such as time range, the InterSensor Service connects to the data source, retrieves observations according to the filter and maps them using the *Observation* class. This means that for every query, relevant observation objects are created dynamically without having any local data storage. It allows encoding observations in common ways for no matter what the data source is. These common observations can then be used further by multiple interfaces for joint analysis and visualizations.

Depending on the sensor type, scenarios, use cases, and applications, sensor observations can be of different data types. For example, a temperature observation is a number while a single observation from a GPS feed is a location. As shown in Figure 4, the Observations class allows encoding observations with different data types and hence providing flexibility to users to encode many possible types of observations.

## 4. Implementing and Configuring the InterSensor Service

The InterSensor Service is a Java application based on the Spring framework [87] and has been released as Open Source software [30]. It includes well-defined classes for each of the mentioned resources. The service can be installed very easily as a standalone application using JAVA JAR commands and can also be deployed on a running server using WAR files.

### 4.1. Adding a Data Source

To work with an InterSensor Service, the first step is to establish a data source connection. The data source connection details can be provided in a configuration file. These configuration files allow defining all the required parameters in order to connect to a specific data source. For example, one publicly available Thingspeak channel is https://thingspeak.com/channels/64242, which can be connected to the InterSensor Service by using the configuration as shown below:
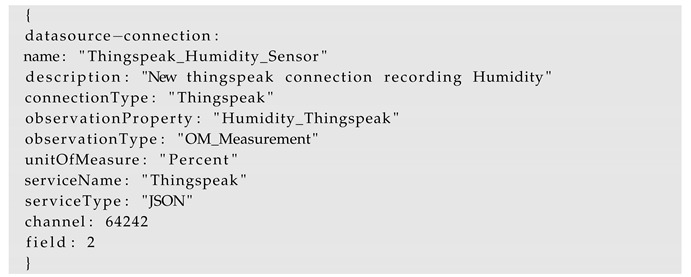


It shows a DHT22 sensor located in Munich, Germany and comprises two observation properties: Field 1 (Temperature) and Field 2 (Humidity). The above-mentioned configuration allows adding a specific property (e.g., Field 2—Humidity) from the Thingspeak channel (with the id 64242) to the InterSensor Service.

Some of the data sources may also require authentication parameters such as username/passwords or an OAuth 2.0 access tokens. The following is an example of a connection to the Twitter API which require the authentication parameters such as *apiKey*, *apiSecret*, *accessToken*, and *accessTokenSecret* in order to retrieve the tweets. The Twitter API supports querying geo-tagged tweets using the geocode parameter. It requires a point location (latitude, longitude) and a radius (e.g., 1 km) around that point. Additionally, even a search keyword can also be provided, however, it can be left blank for retrieving all the tweets. Such parameters can directly be provided in the configuration files.

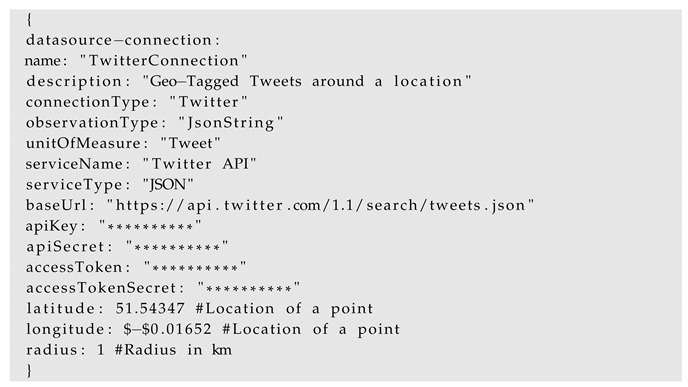


Likewise, connections to arbitrary data sources such as external databases, different IoT platforms (for example, OpenSensors, Weather Underground, SensorThings API) and different file systems such can also be established in easier ways using the pre-defined configuration files. As mentioned in Section 2.3, there might be scenarios where the timeseries data is stored in basic files such as CSV. In these cases, the configuration details can be specified accordingly by providing the file path, and the columns for timestamps and their respective values. Additionally, the information can also be given for other metadata such as the unit of measurement being used and geo-location of the sensor device.

Alternatively, new data sources can be added using an HTTP POST request with the help of any REST client, cURL commands or using software systems such as “HTTP Caller” from the ETL software Feature Manipulation Engine (FME) being very popular in the geospatial domain.

### 4.2. Automated Generation of the Standardized Interfaces

Upon establishing the connection successfully to a data source, the InterSensor Service generates three primary classes *DataSource*, *Timeseries*, and *Observation*. These three classes act as an intermediate layer to connect to a data source, retrieve observations and encode observations “on-the-fly” according to the standardized interfaces OGC SensorThings API and OGC Sensor Observation Service, and the open source Timeseries API. Upon a successful connection, the interfaces for the above-mentioned standards with appropriate classes are automatically generated.

Assuming the server hostname is 127.0.0.1 and the port is 8080, the three classes can be accessed and queried with the help of the following HTTP GET requests:
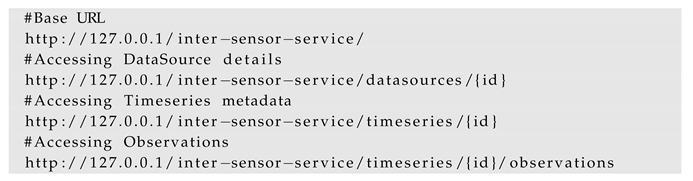


#### 4.2.1. OGC SensorThings API

The SensorThings API comprises of a well-defined data model [21] with different resources such as *Thing*, *Locations*, *Datastream* etc. The InterSensor Service translates the connected data source details according to the SensorThings API data model, which can simply be accessed as follows:
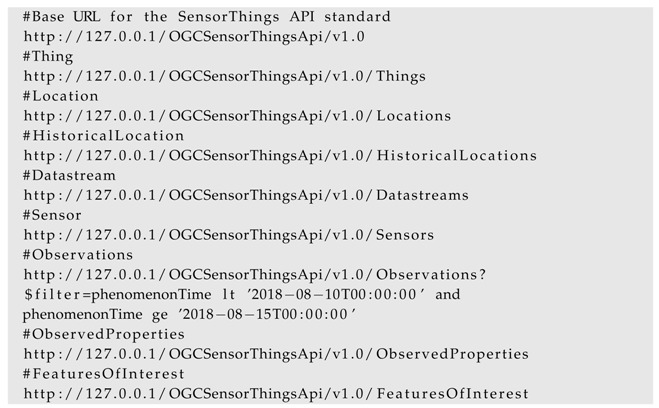


#### 4.2.2. OGC Sensor Observation Service

The Sensor Observation Service (SOS) is a widely adopted web service to query sensor description and metadata and real-time observations. It comprises of well-defined operations such as *DescribeSensor* to retrieve sensor description according to the SensorML standard [18] and *GetObservation* to retrieve real-time observations according to the Observations and Measurements (O&M) standard [19].

For example, the observations from an established InterSensor Service can be queried according to the O&M format by simply using the following *GetObservation* request:
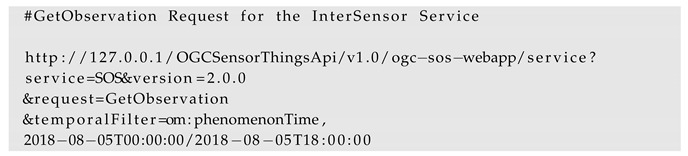


#### 4.2.3. 52° North Timeseries API

The Timeseries API [22] developed by 52° North is a REST-ful web binding to the OGC Sensor Observation Service. While it is not a standard, we decided to support this API because it allows querying and visualizing sensor locations and their observations using the so-called Helgoland Open Source web client [86]. Like the SensorThings API, The Timeseries API also comprises of a well-defined data model and its classes. The observations from an established InterSensor Service can be queried according to the Timeseries API by using the standardized requests:
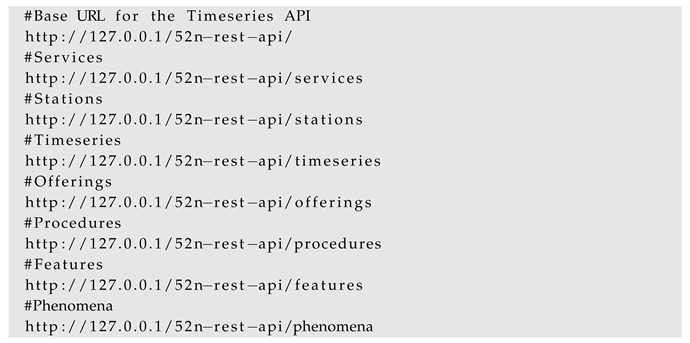


The querying of the data using the standardized requests and responses allow them to be used on the OGC SWE compliant applications. For example, by providing these requests to the Helgoland application, the sensor information can be visualized irrespective of its platform. This is later illustrated in Section 5.2.

## 5. Using the InterSensor Service in Smart City Projects

The InterSensor Service is being employed in the Queen Elizabeth Olympic Park, London under the Smart District Data Infrastructure (SDDI) framework [16]. This project runs within the Smart Sustainable Districts Program [13] of the Climate-KIC of the European Institute for Innovation and Technology (EIT). The SDDI framework allows integrating diverse components such as multiple stakeholders, sensors, IoT devices, simulation tools with a virtual district model representing the physical reality of the district. Within the project, the owners of the district, London Legacy Development Corporation (LLDC), have identified different use cases related to the reduction of resource and energy usage, reduction of waste, reduction of emissions, improvements of well-being, mobility, and in general concerning efficiency.

As shown in Figure 6, for different use cases, the district has access to multiple sensors and IoT devices owned by different stakeholders and partners. For example, two weather stations are located in the park determining the real-time environmental properties such as outside temperature, humidity, wind speed etc. These weather stations are registered with the Weather Underground platform [34]. As a part of the Nature-Smart Cities program (www.naturesmartcities.com), a network of 15 bat monitors is installed across the Olympic Park. The program assumes that bats are considered to be a good indicator species, reflecting the general health of the natural environment. So, a healthy bat population correlates with a healthy biodiversity in the local area. Hence, the smart bat monitors are installed in different habitats across the park and continuously capture data on bat species and activity levels. The observations from these bat monitors are accessed using another platform called OpenSensors [32]. There are smart meters installed in important buildings such as Aquatic Center and Copper Box Arena. These smart meters are used for determining real-time energy consumption (e.g., electricity and gas usage) for the buildings. These meters belong to a company called Engie and are managed within a proprietary platform called C3NTINEL [88]. Similarly, a use case also requires to gather the visitor’s sentiments or experiences by studying the Twitter activity around the park. For this use case, the access to the Twitter API [53] was required to retrieve real-time geo-tagged tweets around the park. For another use case, the park administrators require to assess the impact of scheduled events in the park on the other properties. For example, “if a football match is scheduled in the stadium, what is its impact on the gas consumption of the stadium on that particular day?”. The information of such scheduled events is listed in basic CSV files, which can also be treated as a data source with a timeseries in this context.

As mentioned, these data sources are heterogeneous in nature in a way that they (i) belong to different stakeholders, (ii) are used for different purposes, (iii) based on different platforms and APIs, and (iv) provide different types of observations. However, it is essential to analyze them together for making well-informed decisions. To bring all of them within a common operational framework, the InterSensor Service is being used to connect to all of them. It allows encoding all sensors, their descriptions, metadata and recorded real-time observations using common and mature standards, as well as querying and analyzing them using common interfaces on the OGC Sensor Web Enablement (SWE) compliant applications.

### 5.1. Deployment Options

In the distributed working scenarios having many stakeholders, it is crucial to consider the interests of the stakeholders and types of their platforms before connecting them to the InterSensor Service. It is important to determine whether (i) the platform is open or proprietary, (ii) the platform requires establishing trust by authentication mechanisms, (iii) the stakeholder is willing to share their information to all the users or only to a specific group of users, and so on. For instance, in the case of BIG IoT (as mentioned in Section 2.1), Schmid et al. [89] have defined different integration modes in order to integrate heterogeneous IoT platforms into the BIG IoT ecosystem. These different modes mainly address three major challenges: (i) Interaction of an existing platform with the BIG IoT Marketplace by extending the existing or a new IoT Platform by using the well-defined SDK; (ii) Interaction of the proprietary platforms (with no access to their source code) with the BIG IoT Marketplace using a gateway service, and (iii) Interaction of the constrained device-level platform (e.g., mobile phones or battery-powered sensors) with the BIG IoT Marketplace by using a proxy-service. This proxy-service stores informational resources that are offered by the device-level platform and serve them to the interested consumers upon request.

The InterSensor Service provides several deployment possibilities in a similar way in order to meet the interests of different types of stakeholders. Similar to the Integration Mode 1 as described by Schmid et al. [89], the InterSensor Service allows users to configure a data source connection by extending the existing or a new IoT platform by using the simple Java classes. A medium skilled Java programmer is capable of implementing a new adapter within a day based on the already provided examples. This would allow the user to retrieve sensor observations from all the connected data sources. The provision of the additional standardized interfaces by the InterSensor Service allows users to visualize and analyze the heterogeneous sensor locations and observations within an application in a homogeneous and integrated way as shown in Figure 7a. Such implementations are ideal for scenarios where the involved platforms are open in nature and do not require establishing a trusted, i.e., a secured, connection between the stakeholder and the user.

However, there might be scenarios when a data source e.g., C3NTINEL is proprietary in nature and contains confidential and secure information. In such cases, it is necessary to establish the trust between the stakeholder and the user. The platform requires secure credentials which may be in the form of username/password or OAuth 2.0 access tokens. Due to privacy concerns, the stakeholder would like to avoid revealing the secure credentials to the users of the InterSensor Service. In such cases (similar to the integration mode 2 of the BIG IoT), the respective stakeholder can configure an instance of the InterSensor Service by using the appropriate secure credentials and allow real-time observations to be accessed by the standardized interfaces (as shown in Figure 7b). In this case, without revealing the credentials to a user, the observations can jointly be analyzed with other properties. In this way, it is also possible to configure an additional layer of security facade for providing the appropriate access control. This access control layer allows the stakeholder to configure whether a set of users are allowed to retrieve the observations or not. Such additional security layers can be set up on the OGC-based web services by using an approach proposed by Chaturvedi et al. [90]. In this way, multiple stakeholders can set up the instances of the InterSensor Service for their respective platforms and ensure secure access to their information with the proper access control (see Figure 8).

### 5.2. Joint Visualization and Analysis of Heterogeneous Sensor Platforms and Data Sources

After establishing the connections to multiple heterogeneous data sources, the sensor data and observations could be retrieved according to the external interfaces such as OGC Sensor Observation Service, OGC SensorThings API, and 52° North Timeseries API. This allows applications supporting such OGC SWE interfaces to retrieve the sensor information being retrieved from multiple sensor and IoT platforms.

Figure 9 is a screenshot taken from the Helgoland application developed for visualizing and interacting with sensor data based on the 52° North Timeseries API. The interface from the InterSensor Service can directly be used with the Helgoland application allowing us to interact with observations being retrieved directly from a weather station (outside temperature retrieved from Weather Underground platform), smart meter located in an important building (electricity consumption per minute retrieved from the proprietary C3NTINEL platform) and scheduled events in the same important building (visitor counts during the scheduled event retrieved from a CSV file). Such joint visualization is helpful in determining the correlation between different properties, e.g., ”what is the impact of the weather or any scheduled event on the electricity consumption of a building?”. Of course, such a common standard-based API will also be very valuable for any other kind of application or analysis tool.

### 5.3. Visualization of Sensor Observations with Other OGC Standards

For a different use case in the Queen Elizabeth Olympic Park, it is required to visualize real-time twitter feeds around the park in order to study sentiments and experience of visitors. Using the InterSensor Service, a secure connection to the Twitter API could be established in order to retrieve geo-tagged tweets around the park. The response according to the OGC SWE interfaces makes it suitable to be visualized together with other OGC standards. One such example is shown in Figure 10, where the geo-tagged tweets being retrieved using the InterSensor Service are visualized together with the 3D city objects which are represented according to the OGC CityGML standard. This figure is a screenshot taken from the 3DCityDB-Web-Map application [91] which allows visualizing and interacting with large-scale CityGML-based objects directly within web browsers. The 3DCityDB-Web-Map application is extended for this work to support the OGC SWE interfaces making the application more dynamic. In this way, arbitrary sensor observations can be visualized along with city objects to which they are associated with.

Another similar implementation is done for the city of Augsburg in Germany where the InterSensor Service is used to connect to a proprietary car sharing application. The car sharing application is based on an open interface called the Interface for X-Sharing Information (IXSI) (https://github.com/RWTH-i5-IDSG/ixsi). Using its defined API, it is possible to retrieve information about available rental cars throughout the city in a real-time manner. The InterSensor Service is used to connect to this interface and retrieve the responses according to the OGC SWE interfaces. The standardized response by using the InterSensor Service made it suitable to be visualized along with CityGML-based 3D objects using the 3DCityDB-Web-Map application (see Figure 11).

### 5.4. Performance Evaluation

This sub-section shows the performance of the application by a comparison between the original payload sizes and total response times when directly querying the different platforms and when querying using a standardized interface with the help of the InterSensor Service (as shown in Table 2). In the latter case, payload sizes and response times are calculated for retrieving the observations from an individual data source and encoding them “on-the-fly” according to the OGC SWE interfaces in order to visualize them on an OGC SWE compliant application. For both cases, the mentioned values are the average of five measurements for each data source. However, the performance evaluation for multiple concurrent applications and users is out of scope of this paper and the load tests for this purpose will be performed in the future.

As shown, the observations were retrieved from multiple sources for different use cases: (i) electricity consumption from a proprietary C3NTINEL platform hosted, (ii) Outside Temperature readings from the Weather Underground platform, and (iii) scheduled event and visitor count from a CSV file, (iv) real-time geo-tagged tweets using the Twitter API, and (v) a proprietary car sharing application API. The locations of the running servers are mentioned in the second column. The third column shows the payload sizes and total response times when directly querying the different platforms. The mentioned values are average of five requests that were made against each platform for the time ranges as shown in Figure 9, Figure 10 and Figure 11.

Furthermore, the fourth column shows the values for the requests made against each platform for the same time ranges, however, queried using the standardized interfaces with the help of the InterSensor Service. The payload sizes and the response times, in this case, sum up the requests made to the platform, retrieval of the observations from the data source, encoding the observations “on-the-fly” according to the OGC SWE interfaces and web server latency times. As shown in Figure 9, the observations were retrieved from C3NTINEL platform, Weather Underground platform, and scheduled event and visitor count from a CSV file were encoded according to the OGC Sensor Observation Service (in this paper, according to the 52° North Timeseries API providing a JSON-based RESTful web binding to the SOS in order to be visualized together in the Helgoland web client application). The InterSensor Service and the Helgoland client application are hosted at two different servers located in Technical University of Munich in Germany. Similarly, Figure 10 and Figure 11 show the use cases for connecting to the Twitter API and a proprietary car sharing application API. In this use case, observations were encoded according to the OGC SensorThings API using the InterSensor Service which were used for visualization together with 3D city models on the 3DCityDB-Web-Map application (also hosted in Technical University of Munich).

The last column in Table 2 reflects the latency added by the InterSensor Service in order to retrieve the observations from a data source and encode them “on the-fly” according to the interfaces such as SOS and SensorThings API. The results show that the addition of the InterSensor Service to a sensor platform (or putting the InterSensor Service with a chain of services) adds a latency time up to a few hundred milliseconds. One of the reasons for this additional time is the OGC compliant encodings of original observations from the data sources. The platforms mentioned in the table such as C3NTINEL and Weather Underground provide APIs based on JSON. In this paper, the SWE interfaces (a RESTful web binding of the OGC SOS and the SensorThings API) are also based on JSON, due to which, the payload size and the total response time do not increase significantly compared to the original payload sizes. However, the Observations & Measurements (O&M) encoding of the SOS (which is based on XML encoding and not on JSON encoding) will most likely increase the payload size of the observations compared to the original payload sizes due to difference in encodings. It will be tested in the future. Another determining factor for the additional response time added to the total response time when querying a sensor platform via the InterSensor Service is the network distance. As shown in Table 2, most of the platforms are hosted in different parts of the world such as U.K. and U.S.A. and the InterSensor Service and OGC Web client applications are hosted in Munich (Germany). Since the InterSensor Service is not running on the same machine or same local network as either the original sensor platform or the user application, thus network travel time would also add to the overall response time. Hence, the InterSensor Service ideally should either be running very close to the original sensor platform or to the application (on the same machine or at least in the same local network). If the original platform would be running in China, the InterSensor Service in the U.S. and the application in Europe, then latency would be significantly extended causing performance issues especially in large or real-time observations.

### 5.5. Dealing with Pagination in the Cases of Large Number of Observations

Some of the data sources such as Twitter and SensorThings API provide pagination in order to support efficient retrieval of observations by putting a maximum limit on them. In the case of a request with a very large time period, there might be the possibility of retrieving a large number of observations which may lead to performance issues. The pagination allows dividing the total number of observations in different pages which can be accessed using a cursor. That means it allows retrieving a fixed number of observations (e.g., 100 observations) at one request. The next 100 observations can then be retrieved using a cursor within another request and so on. It makes them suitable to work with lightweight applications and avoid performance issues in case of a very large amount of observations. The InterSensor Service allows dealing with such pagination options while working with different data sources. In the case of the requests with a large time period, the InterSensor service retrieves all the observations by iterating itself to the multiple pages and generate the response according to the respective interface.

## 6. Conclusions and Future Work

This paper describes how interoperability can be established over heterogeneous sensor and IoT platforms and other timeseries data sources using the lightweight InterSensor Service. On the one hand, it allows establishing connections to multiple data sources by using data adapters. On the other hand, it allows querying and visualizing observations from data sources using widely adopted international standards such as the OGC Sensor Observation Service and OGC SensorThings API, and also the open source Timeseries API. In this way, applications and tools can be developed based on these standards without worrying about what different kinds of sensors they use. Multiple sensors can be attached to these infrastructures and their interfaces will always be common for different applications.

The service allows configuring the appropriate security and access controls making it suitable to work with open as well as proprietary data sources in a distributed environment. An important aspect to be considered is the GDPR (General Data Protection Regulation) (https://eugdpr.org/) enabling the InterSensor Service users to view collected personal information. The paper also gives a first performance evaluation by a comparison between the original payload sizes and total response times when directly querying the different platforms and when querying using a standardized interface with the help of the InterSensor Service. However, the performance evaluation for multiple concurrent applications and users is out of scope of this paper and the load tests for this purpose will be performed in the future.

The InterSensor Service is a Java application based on the Spring framework and is available for free as Open Source software (www.intersensorservice.org). The service already supports data adapters for multiple sources such as Thingspeak, OpenSensors, SensorThings, Wunderground and CSV files. In the future, adapters will be developed for other IoT platforms discussed in the paper. Similarly, the support of non-relational databases such as MongoDB and InfluxDB is still an ongoing work. New data adapters can be developed programmatically using the Open Source data model and simple Java classes. A medium skilled Java programmer is capable of implementing a new adapter within a day based on the already provided examples.

The service already supports specific metadata for individual timeseries. Considering observations from heterogeneous data sources, it will be investigated what further metadata will be required. Moreover, current OGC SWE-based standards lack discussions on supporting data sources such as real-time Twitter feeds. It will be discussed with the Standard Working Groups for the respective standards on developing ways to support such virtual sensors as a part of the standard. In the future, the InterSensor Service will also have support for Docker containers to quickly set up instances of the service and move them to the Cloud environment.

## Figures and Tables

**Figure 1 sensors-19-00562-f001:**
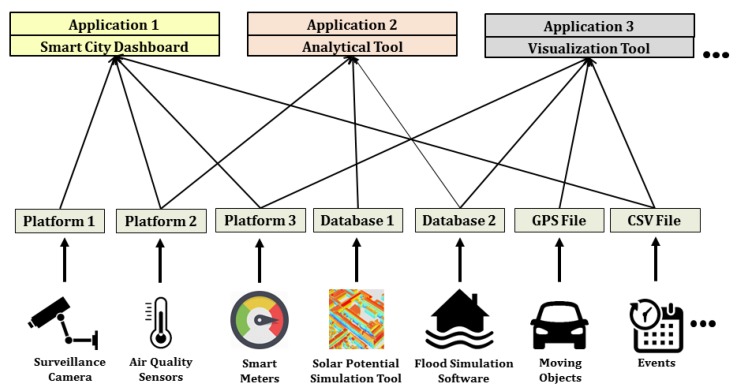
Illustration of heterogeneous data sources for sensor and timeseries data and the requirement to integrate and use them by common applications.

**Figure 2 sensors-19-00562-f002:**
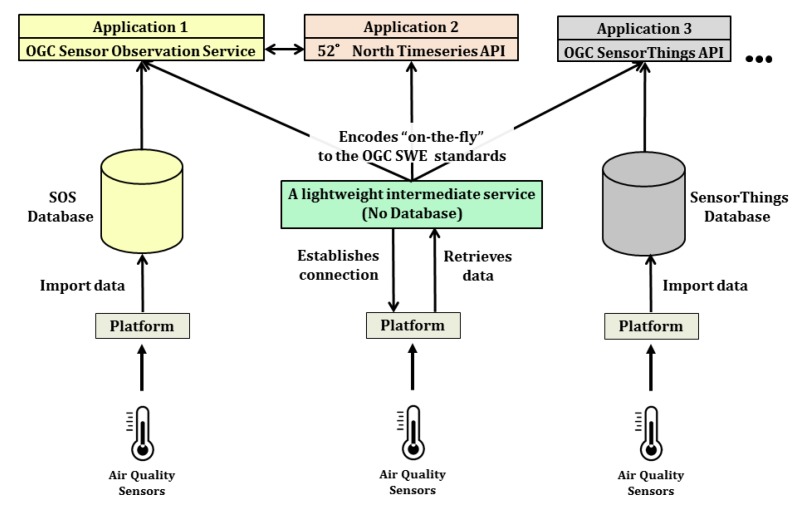
Comparison of the OGC SWE implementations for integrating IoT platforms with them. The implementations of OGC Sensor Observation Service and SensorThings API require importing the observations to their respective data storages. The motivation of this work is a lightweight intermediate service (shown in dark green) allowing connecting to the respective platform, retrieving observations and encoding them “on-the-fly” according to the OGC SWE standardized interfaces.

**Figure 3 sensors-19-00562-f003:**
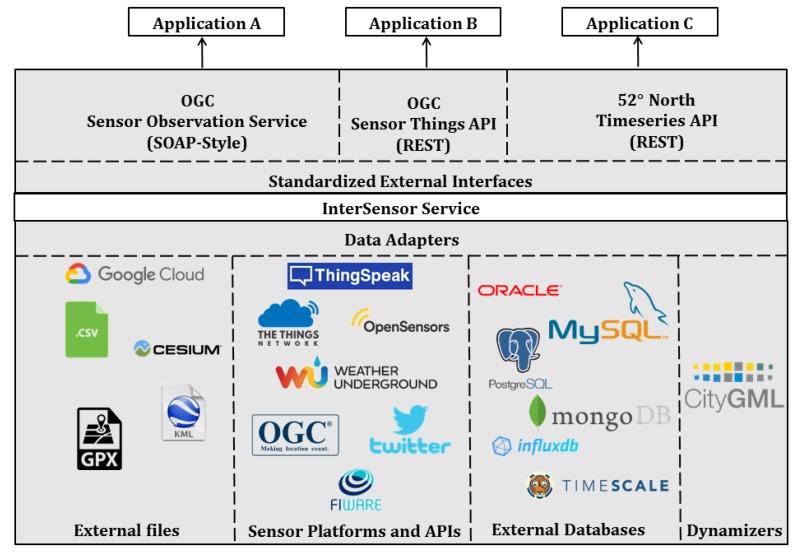
The three-layer architecture of InterSensor Service. The service can be instantiated for individual data sources using adapters and provides standardized external interfaces.

**Figure 4 sensors-19-00562-f004:**
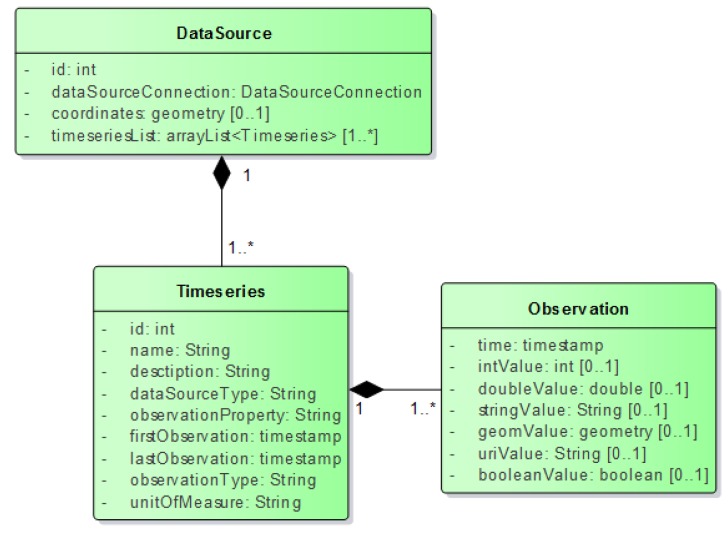
Key resources of InterSensor Service.

**Figure 5 sensors-19-00562-f005:**
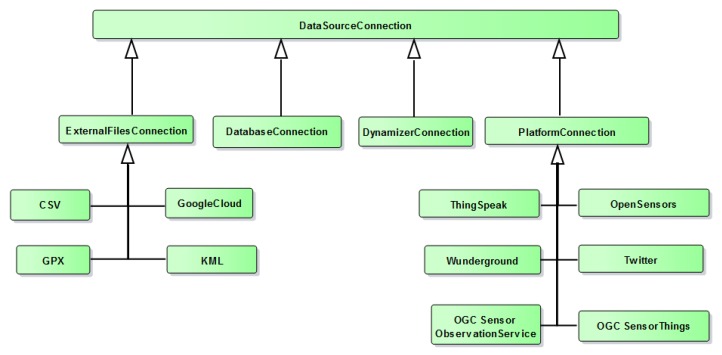
Representation of types of data sources which can be used by the InterSensor Service.

**Figure 6 sensors-19-00562-f006:**
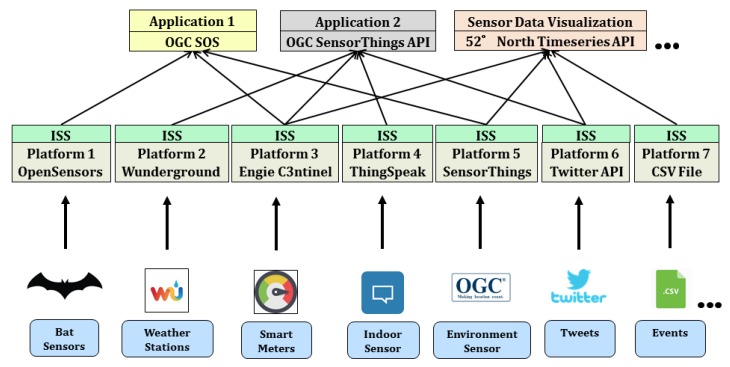
Implementation scenario of the InterSensor Service (ISS) establishing interoperability for different sensor platforms and observations in the district Queen Elizabeth Olympic Park, London.

**Figure 7 sensors-19-00562-f007:**
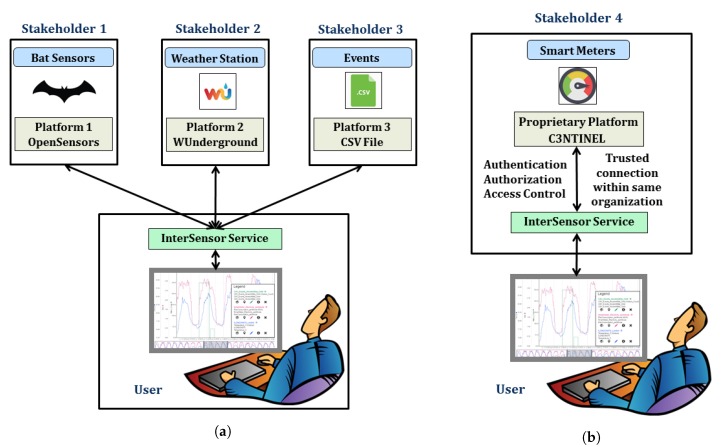
An InterSensor Service can be deployed (**a**) by a user connecting to different data sources, as well as (**b**) by a stakeholder by setting up a trusted connection within same organization.

**Figure 8 sensors-19-00562-f008:**
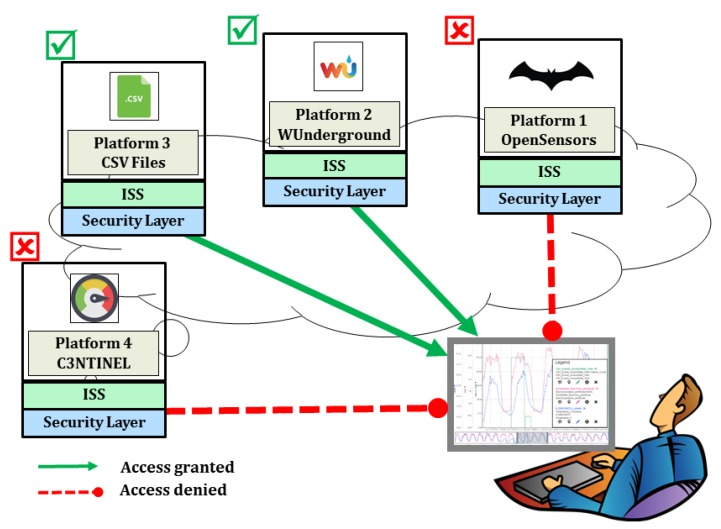
Configuration of the InterSensor Service with an additional security layer by the respective stakeholder. It allows ensuring secure access to the platform with proper access control.

**Figure 9 sensors-19-00562-f009:**
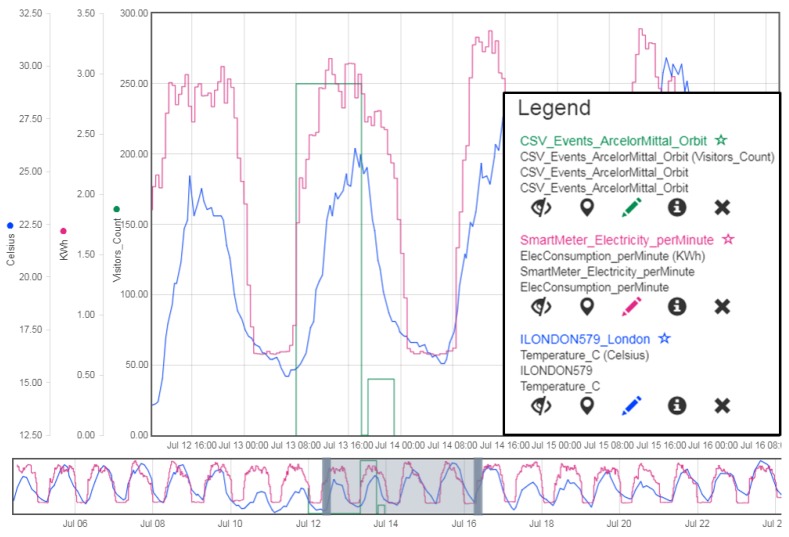
Joint visualizaion of observations being retrieved directly from heterogeneous data sources: (i) electricity consumption from a proprietary C3NTINEL platform in pink, (ii) Outside Temperature readings from the Weather Underground platform in blue, and (iii) scheduled event and visitor count from a CSV file in green. Screenshot taken from the Helgoland web client application.

**Figure 10 sensors-19-00562-f010:**
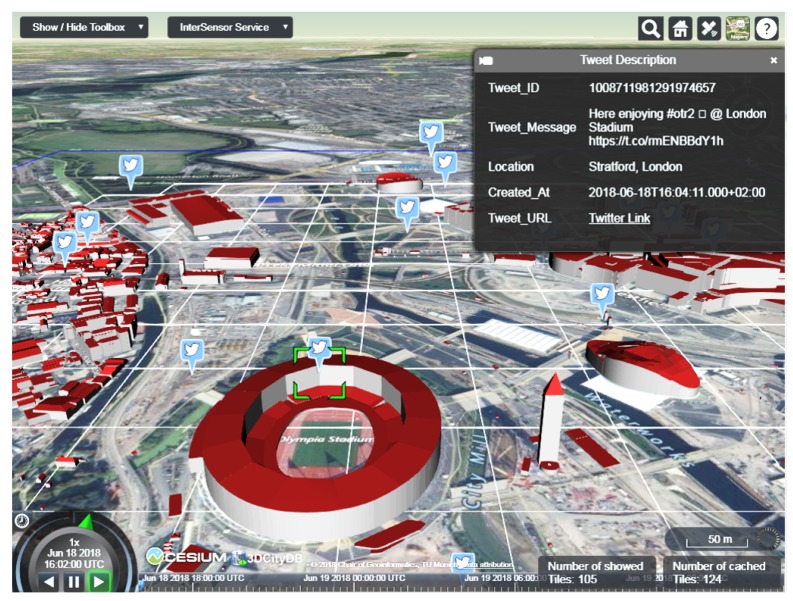
Joint visualization of geo-tagged tweets retrieved by the InterSensor Service along with CityGML-based 3D building objects in the district Queen Elizabeth Olympic Park, London. Screenshot taken from 3DCityDB-Web-Map application [91].

**Figure 11 sensors-19-00562-f011:**
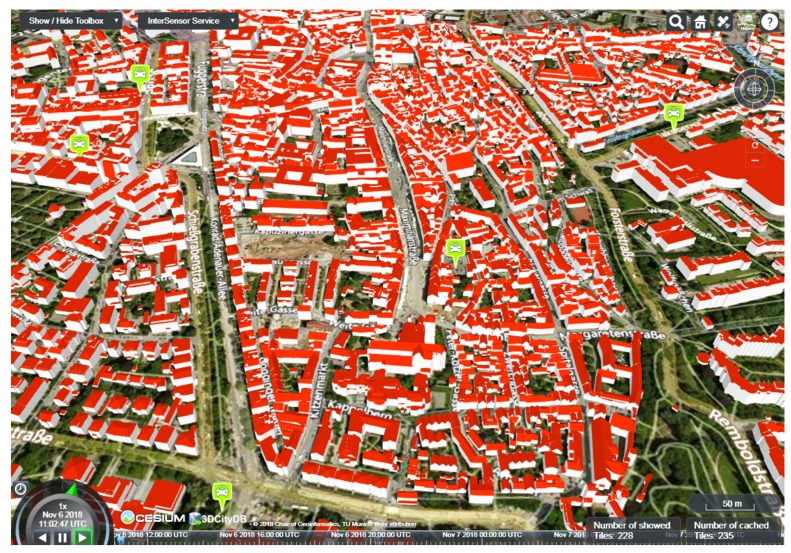
Joint visualization of available rental car information being retrieved by the InterSensor Service along with CityGML-based 3D building objects in the city of Augsburg in Germany. Screenshot taken from 3DCityDB-Web-Map application [91].

**Table 1 sensors-19-00562-t001:** Overview of different types of data sources which are used for storing, managing, and accessing timeseries data for diverse purposes.

Source Type	Description	Examples
Platforms & APIs	(i) allow attaching sensor and IoT devices to them; (ii) allow managing, analyzing, and visualizing real-time observations using sophisticated client applications and well-defined APIs	ThingSpeak [31] OpenSensors [32] The Things Network [33] Weather Underground [34] OGC Sensor Observation Service [20] OGC SensorThings API [21] 52° North Timeseries API [22] FIWARE [26] BIG IoT [25] bIoTope [23] VICINITY [24] symbIoTe [35] Inter-IoT [36] Thingful [37] Smart Emission [38] SenML [39]
Databases	(i) storing and managing time varying observations retrieved from sensor and IoT devices; (ii) managing timeseries values obtained as a result of simulations	Oracle [40] MySQL [41] PostgreSQL [42] TimescaleDB [43] InfluxDB [44] MongoDB [45]
Basic Files	can be used to store timeseries data in structured ways	Comma Separated Values (CSV) Microsoft Excel Sheets
Cloud-based Systems	can be used to store timeseries data in structured ways on the Cloud allowing easy retrieval of the data	Google Fusion Table [46] Google Spreadsheet [47] Microsoft OneDrive [48]
Moving Objects	involve scenarios and applications where the location of an object vary w.r.t. time	GPS Exchange Format (GPX) [49] Keyhole Markup Language (KML) [50] Cesium Language (CZML) [51] Waze API [52]
Social Media	involve real-time social media analytics to be used for behavioral and sentiment analysis	Twitter API [53] Flickr API [54]
Semantic 3D City Models	involve time-dependent and dynamic data from sensors and simulations linked with 3D city objects	CityGML Dynamizers [55]

**Table 2 sensors-19-00562-t002:** Comparison between the original payload sizes and total response times when directly querying the different platforms and when querying using a standardized interface with the help of the InterSensor Service. The payload sizes and total response times shown are average of five requests that were made against each platform for the time ranges as shown in Figure 9, Figure 10 and Figure 11. The InterSensor Sensor Service and OGC web client applications mentioned in this paper are hosted at servers located in the Technical University of Munich, Germany. Locations of the hosted platforms mentioned in the table are determined using www.iplocation.net.

Data Source	Location	Direct Connection		Connection Using ISS			Latency Added by ISS (ms)
		Payload Size (KB)	Total Response Time (ms)	Interface	Payload Size (KB)	Total Response Time (ms)	
C3NTINEL Platform	Plymouth, U.K.	41.0	604	OGC SOS	46.5	837	233
Weather Underground	San Jose, CA, U.S.A.	10.1	884	OGC SOS	13.3	1234	350
CSV File	London, U.K.	0.212	NA	OGC SOS	0.264	387	387
Twitter API	San Francisco, CA, U.S.A.	31.3	765	OGC STA	39.8	1278	513
Car Sharing Platform	Nürnberg, Germany	34.4	779	OGC STA	40.9	1185	406

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
