# Peer review of "Towards Establishing Cross-Platform Interoperability for Sensors in Smart Cities"

_sensors, 2019, doi:10.3390/s19030562_

Reviewer 1 Report

The paper presents the InterSensor service, an open source solution that can stand in the centre of a smart city ecosystem and helps applications to access heterogeneous data sources.

The paper is well written and well structured.

It proposes a solution for a well-known and broadly discussed problem: the gap of (syntactic & semantic) interoperability when dealing with various IoT and sensor data providers.
The background and related work is giving a good overview. From their point of view (geospatial domain),  the authors have understood and captured the state of the art.
The idea of this concept of a data broker/proxy (the InterSensor service) is however not novel.
Nonetheless, the implementation with an innovative data source integration mechanism (via configuration file) seems promising.
The application of the service in a smart city project is nicely described and the screenshots of application vizualizations demonstrate the applicability.
Regarding the different deployment options outlined in Section 5.1, the authors may consider the general "architecture integration modes" proposed by Schmid et al (2017) [1]. Here, a general abstraction of deployments patterns for such a broker/proxy service could be derived and presented. This would make the paper more relevant.
Missing is an evaluation (and discussion) of the approach. At least a short presentation of some quantitative analysis should be added. E.g., how scalable is the service when adding X data sources and having Y numbers of requests per minute, what is the communication overhead (in bytes) compared to a direct communication of the application with the data source?

In conclusion, the paper describes a nice implementation of a data broker/proxy service for the geospatial domain, however, it lacks any sort of evaluation. Hence, I recommend a major revision, in which (at least a short) evaluation is added.

[1] http://link.springer.com/chapter/10.1007/978-3-319-56877-5_3

Author Response

Dear Reviewer,

Thank you very much for your valuable feedback. We would like to confirm that all the suggested changes have been made in the manuscript. The response to the individual feedback is given inline in the attached document.

Thanks and best regards,

The authors

Reviewer 2 Report

The authors present a web service enabling to wrap several data formats (e.g.coming from multiple IoT platforms, files or databases) into the OGC standardized interfaces. In that sense, the authors claim that they provide somehow a meta-platform for managing interoperability between exisiting platforms or standards that was already created for handling this issue, such as projects of the EU ICT-30-2015 programm (Internet of Things and Platforms for Connected Smart Objects). Even though this issue is really relevant and up-to-date, one may wonder whether the proposed work is not only one more platform for doing it.   

Overall, the paper is well presented and written but suffers of several weaknesses that need to be taken into account prior to be considered as accepted: 

* The paper extends the paper already published in the International Smart Cities Conference 2018. This should be at least be stated as such in the introduction (by citing the conference). In addition, it has to be added in a clear and concise way the new contributions of this paper compared to the conference paper

* The scientific contribution/novelty of this paper is not yet clear. It has to be added in the introduction.  

* Projects of the EU ICT-30-2015 program (Internet of Things and Platforms for Connected Smart Objects) seems interesting to be mentioned. That is all the more true that some projects such as bIoTope or Vicinity deal with smart cities or buidings proof-of-concepts. First, the program can be cited when talking about consortia (line 40-41) and those projects could be cited when talking about interoperability issues (e.g. line 68-71). In addition, papers about those projects could be added in the litterature review (in "Platforms & APIs", such as BIG IoT example funded by this same program). It is also mentioned in the litterature review, marketplaces for registering IoT applications. However, only the example of the Big IoT project is stated. What about others marketlaces? such as for instance, bIoTope marketplace (called IoTBnB), or Thingful? Some references that could be considered:

- FRIESS, Peter. Digitising the industry-internet of things connecting the physical, digital and virtual worlds. River Publishers, 2016.

- MYNZHASOVA, Aida, RADOJICIC, Carna, HEINZ, Christopher, et al. Drivers, standards and platforms for the IoT: Towards a digital VICINITY. In : Intelligent Systems Conference (IntelliSys), 2017. IEEE, 2017. p. 170-176.

- ROBERT, Jérémy, KUBLER, Sylvain, KOLBE, Niklas, et al. Open IoT Ecosystem for Enhanced Interoperability in Smart Cities—Example of Métropole De Lyon. Sensors, 2017, vol. 17, no 12, p. 2849.

* No evaluation study was led. Thanks to the authors' proposition, an application can now access to different data sources in an unified way. But what is the cost in terms of response time? Practically, what is the additional time added for encoding "on-the-fly"? If this additional time is in order of milliseconds, it could be fine but if it takes seconds, it could be not suitable for real-time applications (applications with period equals to 1 second are realistic in smart cities). In addition, what about scalability of the service? Since we are talking about IoT applications, how many data sources can be supported by the service in the current use-cases? how many applications/users can be handled by the service? For instance, stress/load tests could be conducted... Note that, as agreed by the authors, the service supports only data adapters for multiple sources. It means that there is no automated way to plug in the service with others data sources, and therefore new data adapters need to be developed. Even if this is entirely understandable, it might be worth precising the cost and difficulty of such developments, in particular in terms of development/efforts time. Is it easy for a new stakeholder to develop his/her own data adapters for a specific needs/data sources?   

* The use-case stated some security and privacy concerns. It is indeed interesting and necessary in such IoT applications. What about the compliance of the InterSensor Service to the EU General Data Protection Regulation (GDPR)?

* Conclusion can be therefore extended with the responses to the two previous points. 

Author Response

Dear Reviewer,

Thank you very much for your valuable feedback. We would like to confirm that all the suggested changes have been made in the manuscript. The response to the individual feedback is given inline in the attached document.

Thanks and best regards,

The authors

Round  2

Reviewer 1 Report

Many thanks for editting the paper according to my reviews.

Well done.

Reviewer 2 Report

The authors took into consideration most of the comments in the review report 1. Even if there is always room for a little further progress (especially in the performance section, the authors put this as future work, what is at least a good thing), I would suggest to accept the paper in the present form.